# Prediction and Analysis of Chlorophyll-a Concentration in the Western Waters of Hong Kong Based on BP Neural Network

Wei-Dong Zhu [1,2,3], Yu-Xiang Kong [1], Nai-Ying He [1,2], Zhen-Ge Qiu [1,2] and Zhi-Gang Lu [4,5,*]

1 School of Marine Science, Shanghai Ocean University, Shanghai 201306, China; wdzhu@shou.edu.cn (W.-D.Z.); m200200644@st.shou.edu.cn (Y.-X.K.); nyhe@shou.edu.cn (N.-Y.H.); zgqiu@shou.edu.cn (Z.-G.Q.)
2 Shanghai Estuary Marine Surveying and Mapping Engineering Technology Research Center, Shanghai 201306, China
3 Key Laboratory of Marine Ecological Monitoring and Restoration Technologies, Shanghai 201306, China
4 School of Resources and Architectural Engineering, Gannan University of Science and Technology, Ganzhou 341000, China
5 Key Laboratory of Mine Geological Disaster Prevention and Control and Ecological Restoration, Ganzhou 341000, China
* Correspondence: 9320060253@gnust.edu.cn

**Abstract:** The Chlorophyll-a (Chl-a) concentration is an important indicator of water environmental conditions; thus, the simultaneous monitoring of large-area water bodies can be realized through the remote sensing-based retrieval of Chl-a concentrations. Together with hyperspectral remote sensing data, a BP neural network model was used to invert chlorophyll-a concentration, with remote sensing reflectance as the input factor. Given the presence of many bands in the hyperspectral data, selecting an appropriate band reflectance as the input factor is crucial to improving inversion accuracy. In this study, a Pearson correlation analysis method was proposed to select bands. A normality test was performed on the reflectance of each band of the Zhuhai-1 hyperspectral remote sensing data, and the significance index was $p < 0.05$. The absolute kurtosis value was less than 10, and the absolute skewness value was less than 3, indicating that the Pearson method was applicable. Pearson correlation analysis was utilised to calculate the correlation coefficient between the reflectance data and chlorophyll-a concentration. Five reflectance data with high correlation were selected as the input factors, and chlorophyll-a concentration was adopted as the output factor. An error backpropagation network model was constructed to predict chlorophyll-a concentration, and a Garson function was added to clarify the connection weights of the input factors in the model construction process. Model 12 was determined as the optimal model on the basis of the criteria of the coefficient of determination, the average relative variance, and the minimum mean square error. The chlorophyll-a concentration was predicted for July and November 2020 in the study area, and the results showed that the predicted values had a small error compared with the measured values. The root-mean-square error and mean relative error of the chlorophyll-a concentration predicted and measured values were 2.12 μg/L and 9.66%, respectively. Significant spatial differences in the Chl-a concentration were observed in the study area due to the influence of islands and land; the Chl-a concentration in July was generally higher than that in November. The results of these studies provide a reference for monitoring the water environment in the study area.

**Keywords:** hyperspectral; chlorophyll-a; BP neural network; spatiotemporal analysis

## 1. Introduction

The waters of Hong Kong are a habitat for marine organisms, such as the Chinese white dolphin. The government of the Hong Kong Special Administrative Region has been committed to protecting the marine environment, which has made the monitoring and management of water quality particularly important. The concentration of chlorophyll-a (Chl-a) represents the highest pigment content in algae, so its concentration is closely related

to the number of algae in the water, making it an indicator for evaluating the eutrophication of water bodies [1]. Traditional sampling and monitoring methods consume extensive manpower and resources. The effective use of historical water quality data to predict the spatial and temporal distributions of Chl-a concentrations can save manpower and resources to a certain extent and improve the efficiency of water quality monitoring. Remote sensing inversion methods used for Chl-a concentration include empirical, semi-analytical, and analytical methods [2]. Empirical methods invert water quality parameters on the basis of statistical relationships between water quality parameters and remote sensing reflectance. In 1983, Carpenter et al. proposed the use of empirical methods to invert Chl-a concentrations in water bodies [3]. Empirical methods can easily build models, but they lack clear physical meanings. Semi-analytical methods select the optimal band or band combination to estimate water quality parameters on the basis of the spectral features of water quality parameters measured by non-imaging spectrometers; then, they find the operational method of sensitive bands. Most studies have considered the radiation transmission of light and the absorption and reflection of substances in water for light, so it has strict physical meanings. This method is widely used in Chl-a concentration inversion. Moses et al. established a three-band model for Chl-a concentration inversion on the basis of MEFZS and MODLES images and estimated the Chl-a concentration in the study area [4]. To further improve the accuracy and universality of inversion models, domestic and foreign scholars have used the bio-optical model of analytical methods to describe the relationship between water quality parameters and the radiance or reflectance spectrum of out-of-water areas. Li et al. inverted the Chl-a concentration of Taihu Lake by establishing an analysis model of water optical transmission in 2004 [5]. To address the complex components of water bodies and adapt to the nonlinear characteristics of research data, Rumelhart and McClelland proposed the basic concept of a backpropagation (BP) neural network in 1986; this network has a strong mapping ability to deal with various complex nonlinear relationships and has achieved great progress in water quality monitoring. In 2019, Wang et al. predicted the water quality of San Francisco Bay by using deep learning methods, that is, through data stability analysis, data cleaning and model establishment, the learning and training of measured water quality data at sampling stations, and predicting the water quality situation of test objects [6]. Zhao et al. applied the BP neural network to the model establishment of the band ratio method for inverting the COD concentration of the Baiyangdian water body [7]. However, the average relative error of the method is 16.5%, and the prediction accuracy is low.

In order to achieve the high precision inversion of chlorophyll-a concentrations across seasons in the coastal waters of western Hong Kong, the spatial and temporal distribution characteristics of chlorophyll-a concentrations in the study waters were obtained. This study adopted the western waters of Hong Kong as the research area and the remote sensing inversion model of Chl-a concentration as the main research object. Actual measurements of Chl-a concentration were conducted in the research area. Since both chlorophyll-a concentration data and remote sensing image data are inter-seasonal and the non-linear characteristics between the data are strong, a BP neural network model was chosen to invert the chlorophyll-a concentration in the study area. On the basis of the hyperspectral remote sensing image data of Zhuhai-1 OHS, the normality of remote sensing reflectance was tested. By using Pearson correlation analysis, sensitive bands were selected as input factors to establish a BP neural network model to predict the Chl-a concentration in the study area. The spatial and temporal distribution characteristics of the Chl-a concentration in the western waters of Hong Kong were then analysed based on the inversion results of the optimal model. This work provides a reference for the monitoring of the Chl-a concentration in the research area and other sea areas.

## 2. Data and Methods

### 2.1. Study Area and Data Sources

The waters off the western coast of Hong Kong are typical coastal waters and are influenced by tropical monsoons with abundant precipitation and notable seasonal differences. The rapid development of the coastal economy and the high degree of industrialisation have led to severe pollution of the water body in the study area by industrial wastewater and residential area sewage, making it a marine region with obvious comprehensive pollution from land resources and marine environments [8–10]. In addition, the water quality control areas, such as Hou Hai Wan and Victoria Harbour, are relatively close, with poor water flow in the sea and an obvious influence from the land topography; thus, the pollutants discharged into them cannot easily flow out. Currently, the water quality type of the coastal waters in western Hong Kong is Class II water, and some areas have mild water body eutrophication. Monitoring the water quality concentration via remote sensing technology has critical implications for balancing the ecological system and restoring the coastal ecological environment.

The actual measured Chl-a concentration data on the western coastal waters of Hong Kong used in this study were downloaded from the Hong Kong Special Administrative Region Environmental Protection Department (https://www.epd.gov.hk/, accessed on 26 January 2022). The data involve the water quality control areas designated by the Hong Kong Environmental Protection Department, including Hou Hai Wan; the northwest, south, and western buffer zones; and Victoria Harbour. The actual Chl-a concentrations were measured in July and November 2020, at a total of 42 sampling points. The sampling point locations are shown in Figure 1, and the actual measured Chl-a concentrations at each sampling point are given in Figure 2.

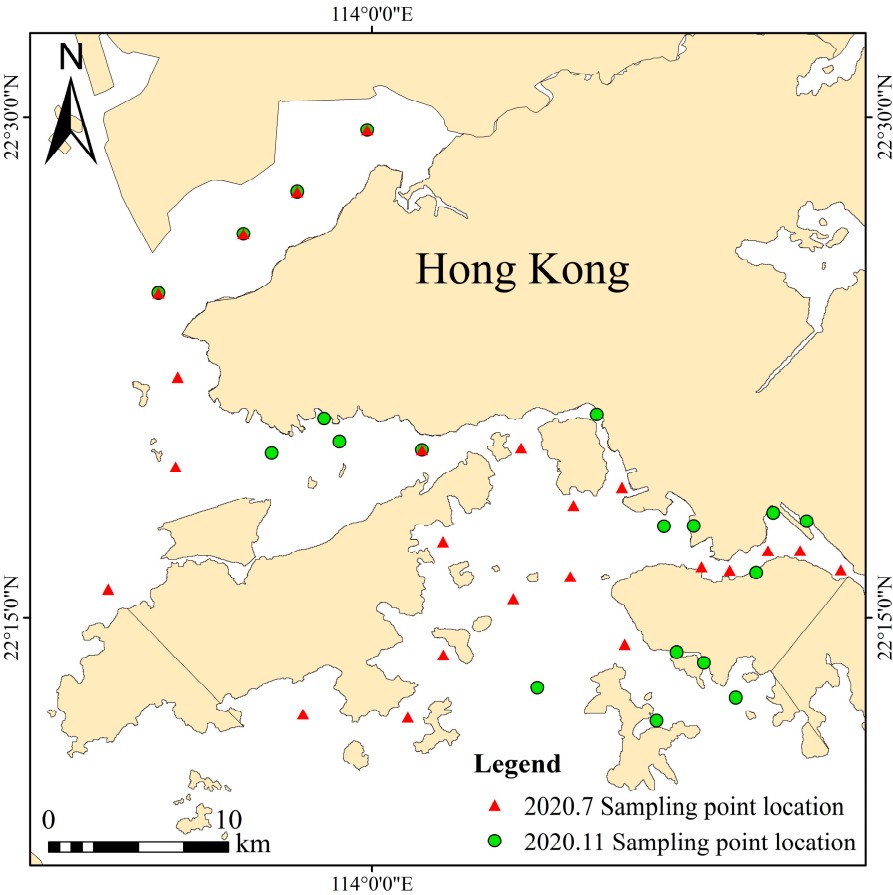

**Figure 1.** Distribution of Sampling Points in the Study Area.

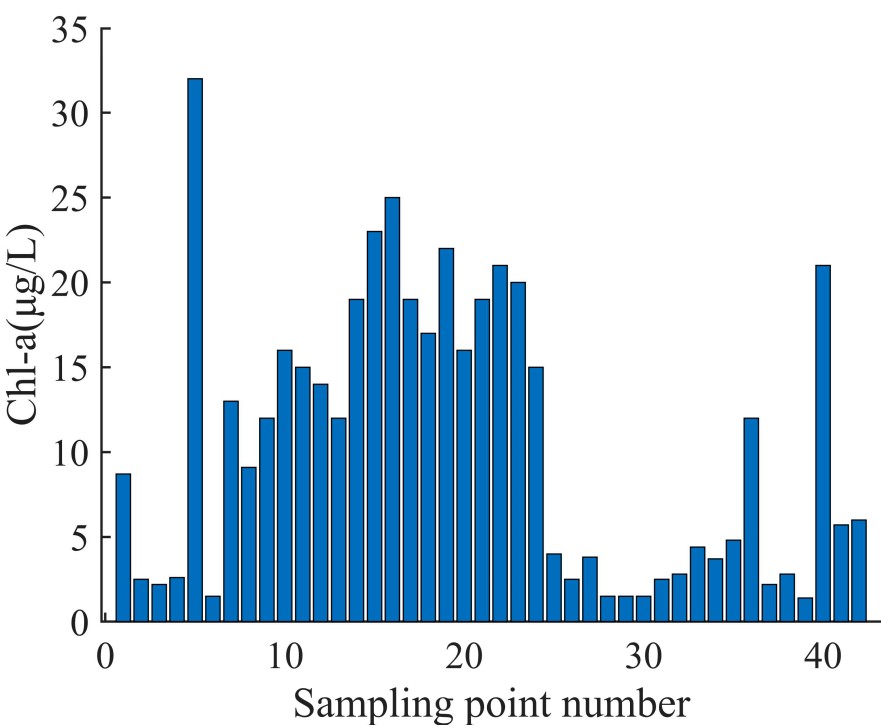

**Figure 2.** Chlorophyll-a Concentration at the Sampling Points.

### 2.2. Pre-Processing of Remote Sensing Images

The hyperspectral satellite images used in this study were obtained from Zhuhai Orbita Aerospace Science and Technology Limited Liability Company. The Zhuhai-1 hyperspectral satellite constellation consists of eight hyperspectral satellites that have stable radiometric performance, low noise, and superior spatial and temporal resolutions, thus providing high-quality data sources for quantitative analysis and information mining in remote sensing [11]. However, due to the relatively short operating time of the Zhuhai-1 satellite, ground-based synchronous experiments for verifying the inversion results are lacking, and the satellite's application in water quality inversion is not widespread. The specific parameters of the satellite are shown in Table 1.

**Table 1.** Parameters of the Zhuhai-1 hyperspectral satellite.

| Parameters | Number |
| --- | --- |
| Spectral range | 400–1000 nm |
| Spectral resolution | 2.5 nm |
| Spatial resolution | 10 m |
| Number of image channels | 32 |
| Orbit altitude | 500 km |
| Imaging range | 1500 km $\times$ 2500 km |

The Zhuhai-1 hyperspectral satellite image has 32 spectral bands, and the centre wavelengths of the blue, green, and red bands are B2 (466 nm), B7 (550 nm), and B12 (640 nm), respectively. The centre wavelengths of each band are shown in Table 2, and the corresponding bands are represented by their channel numbers in this paper.

**Table 2.** Zhuhai-1 hyperspectral band information.

| Channel Number | Wavelength (nm) | Channel Number | Wavelength (nm) | Channel Number | Wavelength (nm) | Channel Number | Wavelength (nm) |
|---|---|---|---|---|---|---|---|
| B1 | 443 | B9 | 580 | B17 | 709 | B25 | 833 |
| B2 | 466 | B10 | 596 | B18 | 730 | B26 | 850 |
| B3 | 490 | B11 | 620 | B19 | 746 | B27 | 865 |
| B4 | 500 | B12 | 640 | B20 | 760 | B28 | 880 |
| B5 | 510 | B13 | 665 | B21 | 776 | B29 | 896 |
| B6 | 531 | B14 | 670 | B22 | 780 | B30 | 910 |
| B7 | 550 | B15 | 686 | B23 | 806 | B31 | 926 |
| B8 | 560 | B16 | 700 | B24 | 820 | B32 | 940 |

High-spectral imagery can receive the radiation information of ground objects through sensors, but the image information is strongly influenced by the sensor and external factors. Therefore, preprocessing remote sensing images is necessary to obtain accurate ground reflection information. Specifically, image cropping, radiometric calibration, atmospheric correction, and uncontrolled point orthorectification are adopted.

### 2.2.1. Radiometric Calibration

Radiometric calibration is the process of converting the raw *DN* values recorded in an image into spectral radiance to eliminate radiometric errors. The conversion formula is as follows:

$$L = \frac{DN}{a} + L_0, \tag{1}$$

where $L$ represents radiance; $DN$ represents the sensor's grayscale value; and $a$ and $L_0$ come from the satellite data file and represent gain and offset values, respectively. In Figure 3, a and b represent the water surface spectral curve before and after radiometric calibration, respectively.

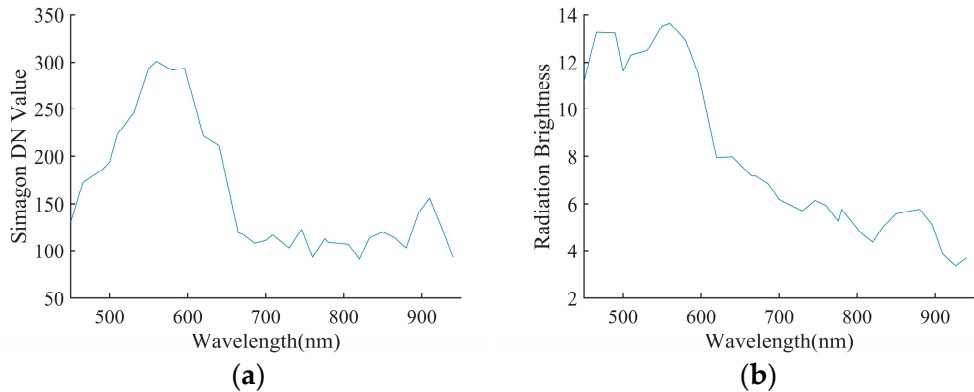

(**a**)            (**b**)

**Figure 3.** Spectral Curves (**a**) Before and (**b**) After Radiometric Calibration.

### 2.2.2. Atmospheric Correction

To remove the interference of atmospheric radiation on ground reflectance information and improve the accuracy of remote sensing data inversion for water quality information, this study used the FLAASH method to perform atmospheric correction on the Zhuhai-1 high-spectral image and applied the MODTRAN4+ model for atmospheric correction [12]. This algorithm has high accuracy and estimates atmospheric properties on the basis of the spectral features of image pixel spectra independently of atmospheric parameter data measured synchronously with remote sensing imaging; it can also effectively remove water vapor and aerosol scattering effects. In Figure 4, a and b represent the spectral feature curve of water before and after atmospheric correction, respectively. Owing to the scattering effects of algae and suspended matter, the corrected spectral curve has the highest reflection

peak at 560–570 nm, a reflection valley on the red band (660 nm), and a secondary reflection peak in the near-infrared band, thus verifying the correctness of atmospheric correction.

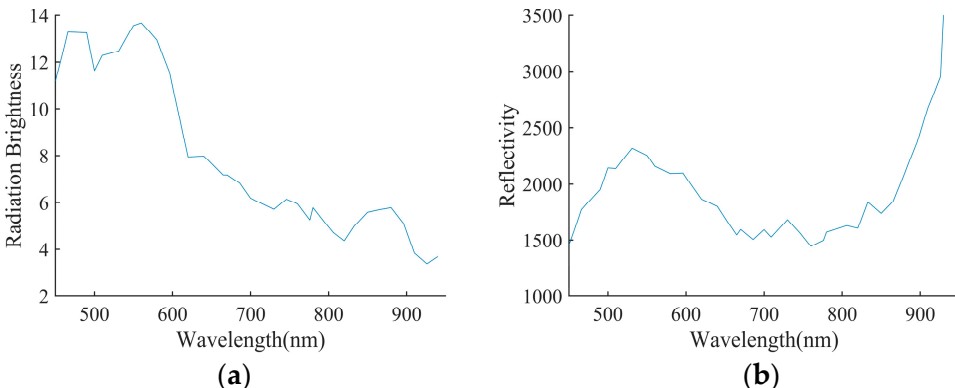

**Figure 4.** Spectral Curves (**a**) Before and (**b**) After Atmospheric Correction.

### 2.2.3. Orthorectification without Ground Control Points

Orthorectification is a process of geometric distortion correction for images, and the main purpose of orthorectification for OHS hyperspectral satellite image data is to correct geometric distortions caused by terrain, camera geometric characteristics, and sensor-related errors. After radiometric calibration and atmospheric correction, RPC information is still retained, and the atmospheric correction results and global DEM can be used directly to perform orthorectification of the image without control points [13].

The reflectance data of each band in the orthorectified remote sensing image were extracted in accordance with the measured coordinates to obtain the reflectance data of each band at the sampling points.

### 2.3. Normality Test of Reflectance Data

Before conducting a correlation analysis, a normality test must be performed to determine whether the remote sensing reflectance data exhibit normality in order to select an appropriate method of calculating the correlation of the research data. In this study, the mean, standard deviation, skewness, and kurtosis of the sample data were calculated using the reflectance data of 32 bands as input variables. The Shapiro–Wilk method was employed to test the normal distribution of the data. This method is a significance hypothesis testing method that compares the sample distribution with the normal distribution from a statistical perspective to determine if the data deviate from or conform to normality. The formula for calculating the test statistic W is as follows:

$$W = \frac{\sum_{i=1}^{n} \left( a_i x_{(i)} \right)^2}{\sum_{i=1}^{n} (x_i - x)^2}, \tag{2}$$

where $x_i$ is the *i*-th order statistic, which is the *i*-th smallest number in the sample, and $x$ is the sample mean. The test results are shown in Table 3.

**Table 3.** Test for normality of distribution.

| Name | Sample Size | Mean | Standard Deviation | Skewness | Kurtosis | Shapiro–Wilk Test Statistic W Value | $p$ |
|---|---|---|---|---|---|---|---|
| B1 | 42 | 845.238 | 651.680 | 0.384 | −1.244 | 0.880 | 0.000 |
| B2 | 42 | 983.810 | 614.002 | 0.108 | −1.594 | 0.864 | 0.000 |
| B3 | 42 | 1058.548 | 588.879 | 0.263 | −1.281 | 0.876 | 0.000 |
| B4 | 42 | 1048.333 | 643.326 | 0.368 | −1.151 | 0.872 | 0.000 |
| B5 | 42 | 1106.976 | 624.713 | 0.473 | −0.916 | 0.881 | 0.000 |
| B6 | 42 | 1159.643 | 664.016 | 0.430 | −1.077 | 0.863 | 0.000 |
| B7 | 42 | 1136.357 | 616.813 | 0.500 | −0.960 | 0.876 | 0.000 |
| B8 | 42 | 1143.952 | 607.887 | 0.519 | −0.914 | 0.880 | 0.000 |
| B9 | 42 | 1108.929 | 610.264 | 0.424 | −1.084 | 0.879 | 0.000 |
| B10 | 42 | 1035.262 | 611.779 | 0.417 | −0.979 | 0.901 | 0.002 |
| B11 | 42 | 1020.262 | 619.634 | 0.235 | −1.356 | 0.882 | 0.000 |
| B12 | 42 | 940.167 | 587.454 | 0.282 | −1.226 | 0.895 | 0.001 |
| B13 | 42 | 900.619 | 530.877 | 0.086 | −1.532 | 0.872 | 0.000 |
| B14 | 42 | 884.857 | 489.228 | 0.138 | −1.341 | 0.894 | 0.001 |
| B15 | 42 | 863.810 | 468.062 | 0.031 | −1.587 | 0.871 | 0.000 |
| B16 | 42 | 901.548 | 513.254 | 0.001 | −1.684 | 0.856 | 0.000 |
| B17 | 42 | 941.310 | 499.818 | −0.073 | −1.698 | 0.850 | 0.000 |
| B18 | 42 | 1000.167 | 649.385 | −0.198 | −1.913 | 0.785 | 0.000 |
| B19 | 42 | 846.452 | 525.086 | 0.136 | −1.527 | 0.888 | 0.001 |
| B20 | 42 | 876.381 | 539.786 | 0.229 | −1.284 | 0.922 | 0.007 |
| B21 | 42 | 896.643 | 541.395 | 0.047 | −1.623 | 0.876 | 0.000 |
| B22 | 42 | 933.714 | 569.613 | 0.070 | −1.604 | 0.882 | 0.000 |
| B23 | 42 | 1000.690 | 571.033 | −0.002 | −1.621 | 0.882 | 0.000 |
| B24 | 42 | 1060.952 | 608.497 | −0.082 | −1.788 | 0.843 | 0.000 |
| B25 | 42 | 1207.952 | 694.472 | −0.214 | −1.970 | 0.747 | 0.000 |
| B26 | 42 | 1196.690 | 667.880 | −0.238 | −1.958 | 0.740 | 0.000 |
| B27 | 42 | 1239.571 | 656.861 | −0.250 | −1.941 | 0.742 | 0.000 |
| B28 | 42 | 1458.833 | 732.599 | −0.257 | −1.926 | 0.749 | 0.000 |
| B29 | 42 | 1617.548 | 779.243 | −0.197 | −1.933 | 0.775 | 0.000 |
| B30 | 42 | 1766.619 | 839.419 | −0.086 | −1.776 | 0.839 | 0.000 |
| B31 | 42 | 1866.286 | 858.027 | −0.009 | −1.640 | 0.879 | 0.000 |
| B32 | 42 | 2908.024 | 1294.357 | 0.729 | 0.322 | 0.951 | 0.068 |

As shown in the table, except for the statistical $p$ value of B32, which is greater than 0.05, the $p$ values of all other bands are less than 0.05, indicating that, except for B32, the data of each band present significance but do not exhibit the trait of normal distribution. However, the normality test has strict requirements, which are often difficult to meet. If the absolute kurtosis value is less than 10 and the absolute skewness value is less than 3, the data can be considered normally distributed, although not strictly normal [14,15]. Therefore, Spearman correlation analysis is unsuitable for studying the data. This study used Pearson correlation analysis to calculate the correlation between Chl-a concentrations and remote sensing reflectance.

*2.4. Pearson Correlation Analysis*

On the basis of the normality test results, a Pearson correlation analysis was performed to calculate the correlation between each spectral band and a combination of spectral reflectance with Chl-a concentration. The formula is defined as follows:

$$\rho = \frac{\frac{1}{n}\sum_{i=1}^{n}\left(R(x_i) - \overline{R(x)}\right) * \left(R(y_i) - \overline{R(y)}\right)}{\sqrt{\left(\frac{1}{n}\sum_{i=1}^{n}\left(R(x_i) - \overline{R(x)}\right)^2\right) * \left(\frac{1}{n}\sum_{i=1}^{n}\left(R(y_i) - \overline{R(y)}\right)^2\right)}}, \quad (3)$$

where $n$ is the total number of samples and $x_i$ and $y_i$ are the values of the $i$-th sample included in the calculation. The correlation coefficients between spectral reflectance and Chl-a concentration for each spectral band are shown in Figure 5.

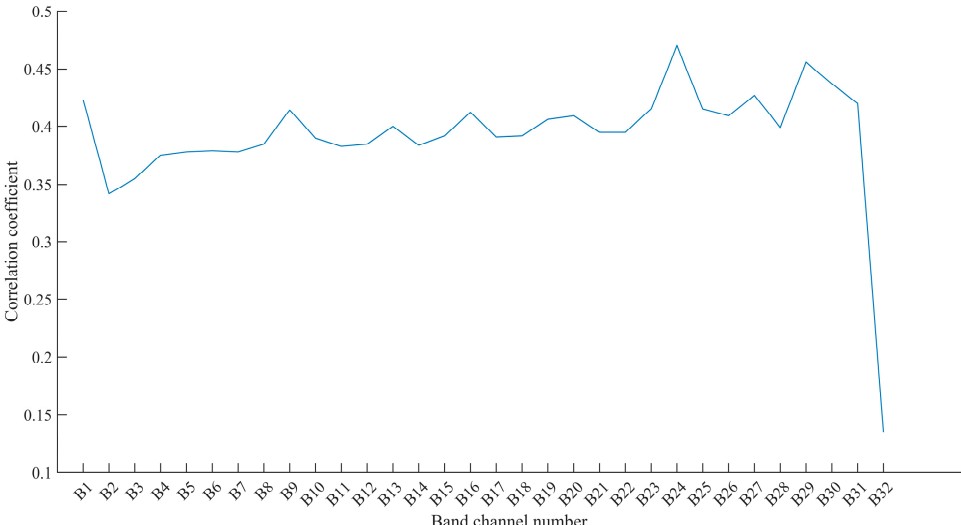

**Figure 5.** Single-band Correlation Coefficient.

If the single band with the highest correlation coefficient (B27) is selected as the independent variable to construct the Chl-a concentration inversion model, the determination coefficient ($R^2$) of the linear fitting equation is only 0.35, which makes it difficult to achieve effective inversion of the Chl-a concentration in the study water area. Therefore, the reflectance of all bands must be combined based on the Pearson correlation analysis to improve the inversion accuracy [16]. Band combination is based on arithmetic operations, and due to the large number of bands in hyperspectral images, the combination is highly complex. In the combination, band ratio is the main method, and other methods, such as logarithmic transformation, are used as auxiliary operations, resulting in a total of 6000 correlation coefficients. Bands with Pearson correlation coefficients greater than 0.7 were selected in this study to build the BP neural network model.

### 2.5. BP Neural Network Model Construction

The BP network is trained by the error BP algorithm [17]. Its basic processing unit is a nonlinear input–output relationship. The input and output values can change continuously, and its activation function is an exponential S-shaped function (sigmoid), which is a differentiable, non-decreasing, nonlinear, continuous function. The expression is as follows:

$$F(x) = \frac{1}{1+e^{-x}} \tag{4}$$

The BP neural network mainly consists of input, hidden, and output layers. On the basis of the input factors determined by grey relational analysis, the basic structure of the neural network model in this experiment is shown in Figure 6. For ease of computation in the neural network model, the input factors are denoted as B1–B5 from top to bottom. The specific values of the input factors are shown in Table 4.

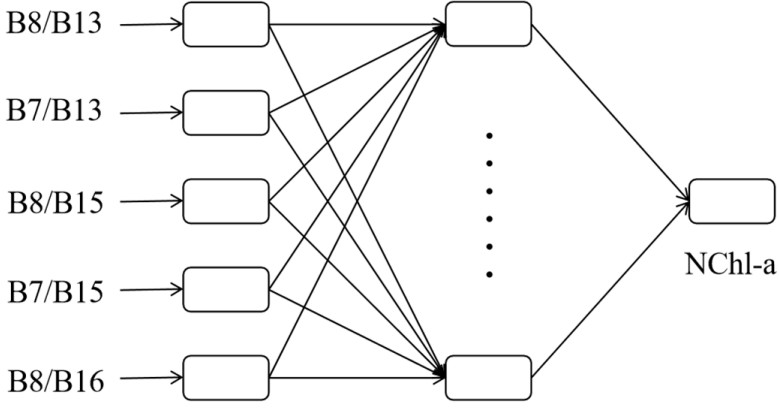

**Figure 6.** BP Neural Network Architecture Diagram.

**Table 4.** Input factors for BP neural network models.

| B8/B13 | B7/B13 | B8/B15 | B7/B15 | B8/B16 |
|---|---|---|---|---|
| 1.24766098 | 1.442111959 | 1.265272427 | 1.481314433 | 1.462468193 |
| 1.256030702 | 1.358032009 | 1.256578947 | 1.364285714 | 1.358624778 |
| 1.287246722 | 1.396250808 | 1.296781883 | 1.444887118 | 1.406593407 |
| 1.413385827 | 1.393272962 | 1.346456693 | 1.512159175 | 1.327296248 |
| 1.067746686 | 1.011157601 | 1.122974963 | 1.193270736 | 1.063458856 |
| 1.393134715 | 1.363117871 | 1.364637306 | 1.410307898 | 1.335234474 |
| 1.332112332 | 1.315250151 | 1.373015873 | 1.523712737 | 1.355635925 |
| 1.382597569 | 1.339739616 | 1.373640435 | 1.403267974 | 1.331060136 |
| 1.148760331 | 1.152570481 | 1.155371901 | 1.254937163 | 1.15920398 |
| 1.182758621 | 1.111831442 | 1.161206897 | 1.171304348 | 1.091572123 |
| 1.150961538 | 1.08032491 | 1.139423077 | 1.159491194 | 1.069494585 |
| 1.151823579 | 1.183958152 | 1.199321459 | 1.283121597 | 1.232781168 |
| 1.162672476 | 1.141126158 | 1.14379085 | 1.214340786 | 1.12259444 |
| 1.068457539 | 1.071242398 | 1.056325823 | 1.067425569 | 1.059079062 |
| 1.124132614 | 1.191176471 | 1.108712413 | 1.2417962 | 1.174836601 |
| 1.135433071 | 1.199667221 | 1.146456693 | 1.1375 | 1.211314476 |
| 1.108130763 | 1.024012393 | 1.148365465 | 1.164965986 | 1.061192874 |
| 1.02617801 | 1.129682997 | 1.036649215 | 1.091911765 | 1.141210375 |
| 1.4425 | 1.397094431 | 1.4275 | 1.266075388 | 1.382566586 |
| 1.491094148 | 1.468671679 | 1.582697201 | 1.382222222 | 1.558897243 |
| 1.750629723 | 1.786632391 | 1.672544081 | 1.588516746 | 1.706940874 |
| 1.902140673 | 2.046052632 | 1.862385321 | 1.891304348 | 2.003289474 |
| 1.857142857 | 1.63368984 | 1.832826748 | 1.838414634 | 1.612299465 |
| 1.867507886 | 1.81595092 | 1.958990536 | 1.84272997 | 1.904907975 |
| 1.198630137 | 1.209677419 | 1.308219178 | 1.234913793 | 1.320276498 |
| 1.076419214 | 1.117913832 | 1.203056769 | 1.249433107 | 1.249433107 |
| 1.654411765 | 1.642335766 | 1.672794118 | 1.666666667 | 1.660583942 |
| 1.26910299 | 1.201257862 | 1.445182724 | 1.306306306 | 1.367924528 |
| 1.762711864 | 1.787965616 | 1.84180791 | 1.77173913 | 1.868194842 |
| 1.695906433 | 1.779141104 | 1.666666667 | 1.592178771 | 1.748466258 |
| 1.655870445 | 1.649193548 | 1.599190283 | 1.573705179 | 1.592741935 |
| 1.923423423 | 1.66796875 | 1.918918919 | 1.516014235 | 1.6640625 |
| 1.98206278 | 1.655430712 | 2.085201794 | 1.631578947 | 1.741573034 |
| 1.792253521 | 1.647249191 | 1.971830986 | 1.806451613 | 1.812297735 |

The main learning process of the BP neural network can be divided into the forward propagation and error backpropagation processes. The forward propagation process can

be simply understood as the input and output process of the sample. The experimental sample enters from the input layer and reaches the hidden layer. Its function is to generate a set of output values through a threshold-type activation function and then transmit the output values to the connected output layer units [18]. The error backpropagation process is when the error between the calculated value of the network and the expected value does not meet the requirements; that is, when the output error is too large, the neural network propagates the error layer by layer in a backward direction from the input layer to the output layer and continuously adjusts the corresponding weights during the propagation process. The learning process of the BP neural network is iterative until the output error meets the requirements.

Seventy percent of the water quality data without validation samples were randomly selected as the training sample set for the model, and the remaining thirty percent were used as the testing sample set. Given that a complex network structure is not conducive to improving the model's fitting accuracy, a three-layer BP neural network structure can make the learning efficiency reach a high value. Therefore, a single-hidden-layer three-layer BP neural network was used in this work [19]. The input factor had five dimensions, the output layer was the normalised concentration value of Chl-a, and the maximum training batch was 100. The number of hidden layer neurons was increased from 1 to 20 one by one, and the accuracy of the obtained 20 neural network models was evaluated. The most suitable BP neural network model for studying the water area was then selected.

After obtaining 20 different neural network models with different numbers of hidden layer neurons, model selection was conducted to determine the optimal model. The accuracy indicators for model selection were the coefficient of determination ($R^2$), average relative variance (ARV), and mean square error (MSE) of each model. $R^2$ is a commonly used indicator to evaluate the fitting accuracy of a model, and the value range of $R^2$ is (0, 1). The closer $R^2$ is to 1, the higher the accuracy of the model [20]. The degree of explanation of the independent variable to the dependent variable increases naturally as the goodness of fit improves, and its function definition is as follows:

$$R^2 = 1 - \frac{SS_{res}}{SS_{tot}} = \frac{SS_{reg}}{SS_{tot}}, \quad (5)$$

where $SS_{res}$, $SS_{reg}$, and $SS_{tot}$ represent the regression sum of squares, residual sum of squares, and total sum of squares, respectively.

The overall performance of a neural network in practical applications, which is called its generalisation ability, must be evaluated. This ability reflects the neural network's adaptability to new data that have not appeared in the training set. If the predicted and measured values differ only minimally, the model is relatively ideal. ARV is usually used to measure the difference between the predicted and measured values [21]. Its function definition is shown in Equation (6). The value range of ARV is [0, 1]. The smaller the average relative variance is, the better the prediction accuracy of the model. ARV = 0 indicates that the model has achieved the expected prediction effect. Conversely, ARV = 1 means that the model has only achieved the average prediction effect. Meanwhile, MSE is used as a criterion for judging the optimal model. The smaller MSE is, the more accurate the prediction result of the BP neural network. Its function definition is shown in Equation (7).

$$ARV = \frac{\sum_{i=1}^{n}(x_i - x_i)^2}{\sum_{i=1}^{n}(x_i - \hat{x}_i)^2}, \quad (6)$$

$$MSE = \frac{\sum_{i}^{n}(x_i - x_i)^2}{n}, \quad (7)$$

## 3. Results and Analysis

### 3.1. Determination of the Number of Hidden Layer Neurons

The selection of the BP neural network model mainly relied on the comprehensive consideration of three indicators: $R^2$, ARV, and MSE. As the number of hidden layer neurons increased, different neural network models with hidden layer node numbers ranging from 1 to 20 were generated during the BP neural network operation. Each model was calculated 100 times, and the values of the three indicators were obtained and presented in the form of box plots.

As shown in Figure 7, the $R^2$ values of the various models were concentrated between 0.5 and 0.6, and the fitting results were relatively stable. The determination coefficient fluctuated slightly with the presence of hidden layer neurons, and Model 12 had the highest determination coefficient. The ARV values fluctuated around 0.5, and only a small difference was observed between the models. The MSE values were all between 0.03 and 0.04, and their overall trend of change (i.e., showing slight fluctuations with the increase in neurons) was similar to that of ARV. According to the distribution chart, Model 12 performed better than the other models on the whole. Therefore, considering the overall trend of the numerical changes and the degree of numerical discreteness, Model 12 was determined as the optimal model of the BP neural network. The input factor for Model 12 is the five band reflectance ratios, B8/B13, B7/B13, B8/B15, B7/B15, and B8/B16; the number of neurons in the hidden layer is 12; and the output layer is the chlorophyll-a concentration.

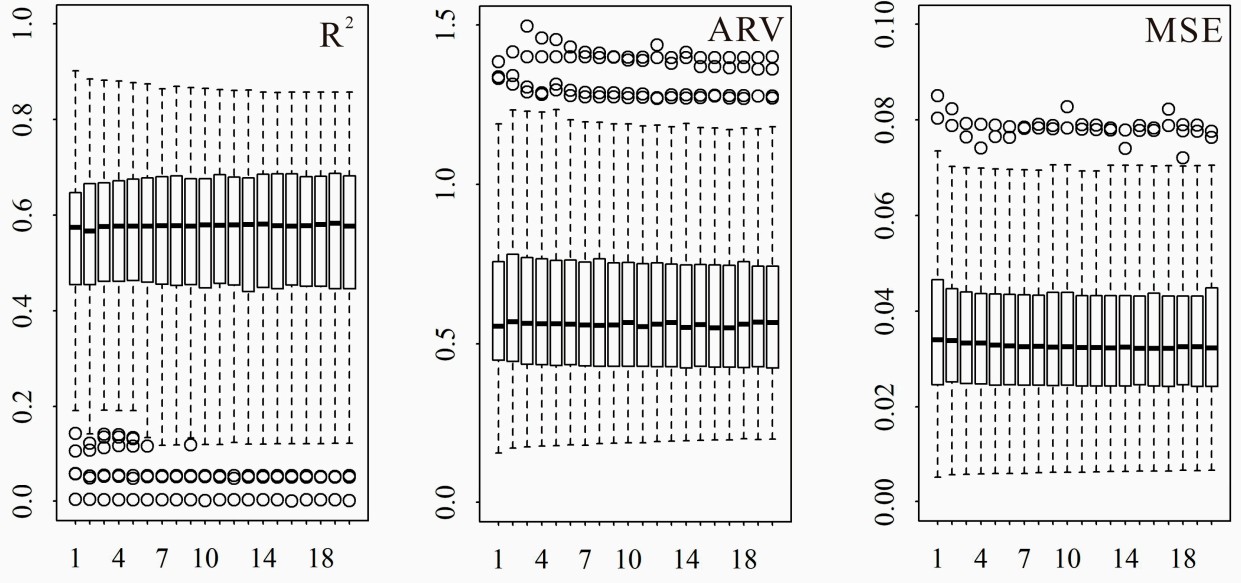

**Figure 7.** Distribution of the $R^2$, MSE, and ARV of All Models.

### 3.2. Analysis of Input Factor Connection Weights

Neural interpretation diagrams (NIDs) were constructed for Model 12, and straight lines were used to represent the connection weights between the input layer and the hidden layer and between the hidden layer and the output layer. The Garson algorithm was employed to analyse the sensitivity of the neural network based on connection weights, that is, to obtain the sensitivity of the weights through the neural network and use the product of the connection weights to obtain the importance of the input variables to the output variables (i.e., relative contribution value), which provides a reference for the improvement of subsequent models [22,23]. The neural interpretation diagram of the western waters of Hong Kong is shown in Figure 8.

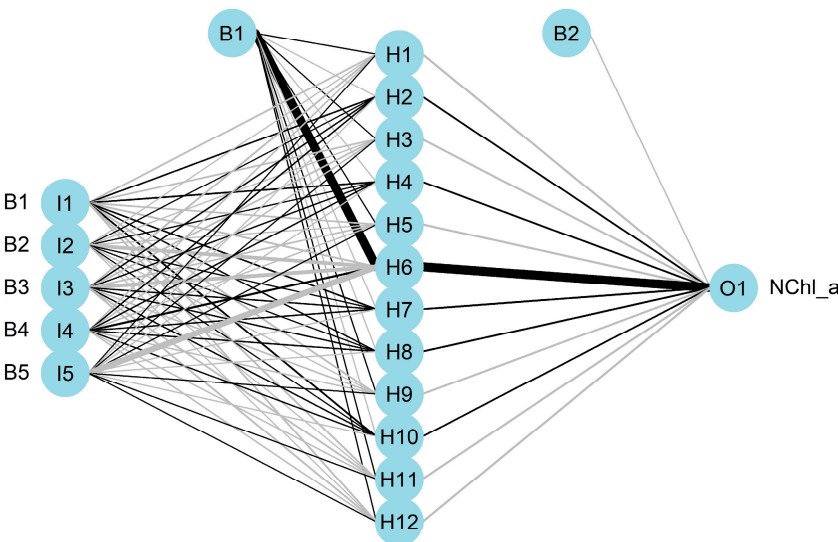

**Figure 8.** Neural Network Interpretation of Model 12.

In the neural interpretation diagram, B1 and B2 represent the biases added at each step, the black connecting lines represent positive signals, the thickness of the lines represents the degree of correlation between the input factors, and the grey connecting lines represent negative signals.

Given the dense connection lines in the neural interpretation diagram, observing the weight status of each input factor is inconvenient. Therefore, the Garson algorithm was used to generate a histogram of input factor connection weights, as shown in Figure 9. The relative contribution values of each input variable to the output variable in the model construction process were obtained under the condition that the number of hidden layer neurons was 12. The purpose was to clearly view the connection weight status of each input factor in the model construction process.

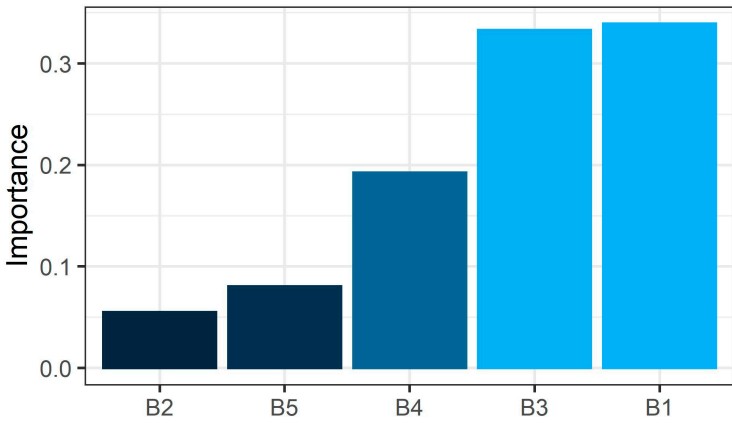

**Figure 9.** Garson Diagram of Model 12.

### 3.3. Model Accuracy Verification and Evaluation

On the basis of the Zhuhai-1 remote sensing images, a BP neural network model was constructed to predict the Chl-a concentration in the western waters of Hong Kong. The relative error (RE), root mean square error (RMSE), and mean relative error (MRE) are used as accuracy evaluation indicators, and their calculation formulae are, respectively, as follows.

$$RE = \frac{C_i - A_i}{C_i} 100\%,$$

(8)

$$RMSE = \frac{\sum_{i=1}^{n}(A_i - C_i)^2}{n}, \qquad (9)$$

$$MRE = \frac{100\%}{n}\sum_{i=1}^{n}\left|\frac{C_i - A_i}{C_i}\right|, \qquad (10)$$

The relative error (RE), root-mean-square error (RMSE), and MRE between the predicted and measured values are shown in Table 5, and the distribution of the predicted and measured Chl-a concentrations is given in Figure 10.

**Table 5.** Model accuracy validation.

| Measured Concentration (μg/L) | Predicted Concentration (μg/L) | Relative Error (%) | Root-Mean-Square Error (μg/L) | Mean Relative Error (%) |
|---|---|---|---|---|
| 15 | 14.53 | 3.10 | | |
| 19 | 18.41 | 3.06 | | |
| 23 | 21.87 | 4.90 | | |
| 19 | 23.14 | 21.80 | | |
| 19 | 18.04 | 5.02 | 2.12 | 9.66 |
| 15 | 14.52 | 3.19 | | |
| 4.8 | 5.64 | 17.70 | | |
| 21 | 17.10 | 18.52 | | |

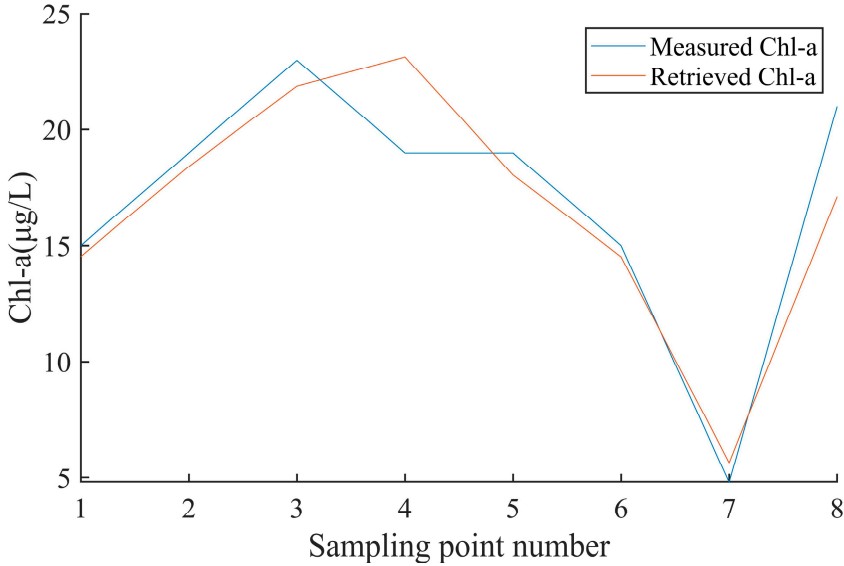

**Figure 10.** Retrieval Results of the BP Neural Network Model and the Measured Values.

As shown in the table, the RMSE and MRE between the predicted and measured Chl-a concentrations were 2.12 μg/L and 9.66%, respectively. Amongst the eight validation sites, five had REs below 10%, and one had an RE above 20%. The minimum and maximum REs were 3.10% and 21.80%, respectively, with the maximum RE occurring in the northwest water quality control area. As shown in Figure 10, the predicted Chl-a concentrations were generally consistent with the measured values, indicating the high overall prediction accuracy of the model.

### 3.4. Spatiotemporal Analysis of Chl-a Concentration in the Study Area

On the basis of the BP neural network model, the predicted Chl-a concentrations at each sampling site in the study area were obtained, and Chl-a concentration distribution maps for July and November 2020 were plotted using ArcGIS 10.8 software, as shown in Figure 11.

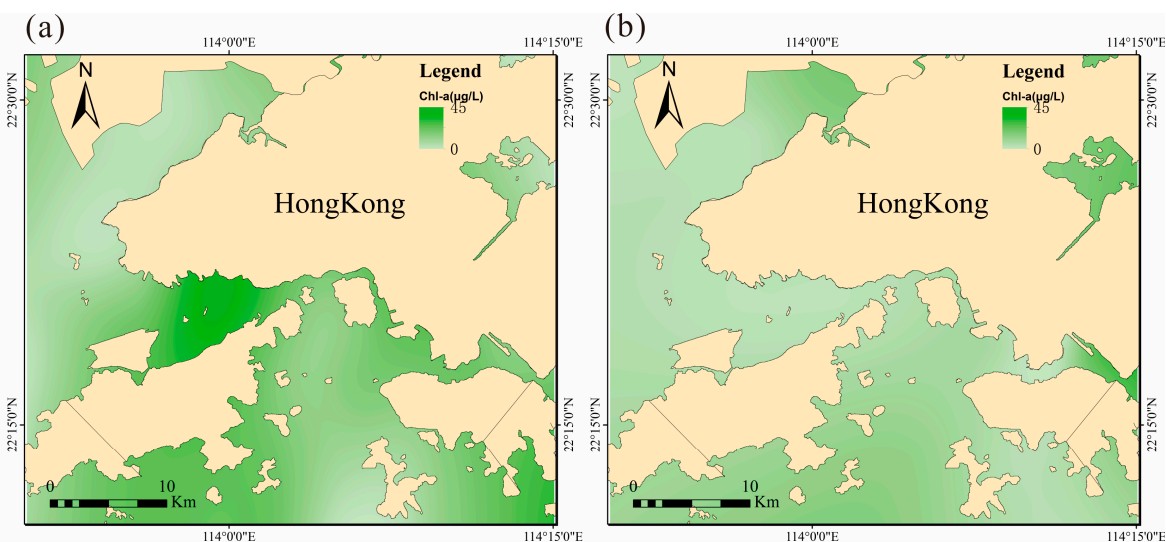

**Figure 11.** Chlorophyll-a Concentration Inversion Map of the Western Waters Near Hong Kong: (**a**) Chl-a Retrieval Concentration Distribution Map Obtained for July 2020 and (**b**) Chl-a Retrieval Concentration Distribution Map Obtained for November 2020.

Significant spatial differences in the Chl-a concentration were observed in the study area due to the influence of islands and land. In the northern Houhai Bay water quality control area, the Chl-a concentration remained at a low level due to the implementation of water environmental management work, with only a small high-concentration area appearing in the eastern bay in November with an average concentration of 15 μg/L. In addition, high Chl-a concentrations were mainly observed in eastern Victoria Harbour, the northern part of Lantau Island, and some narrow nearshore areas in the east; the spatial distribution of the Chl-a concentration was considerably limited by topography. From a temporal perspective, the Chl-a concentration in this sea area showed obvious seasonal variations, with higher concentrations in summer than in autumn and winter, mainly due to the temperature-induced proliferation of algae and plankton in the water. Therefore, the Chl-a concentration in July was generally higher than that in November. Moreover, the high Chl-a concentration values at the individual sites were attributed to the special conditions occurring in small, enclosed areas caused by the increase in inorganic salts due to pollutant discharge. This feature was also reflected by the predicted values, indicating that the network could accurately predict the spatiotemporal trend of the Chl-a concentration.

## 4. Discussion

### 4.1. Normality Test

In the construction process of a BP neural network model, the accuracy of the model prediction depends on the strength of the correlation between the input and output factors. Selecting highly correlated reflectance data as the input factor can effectively improve the model's fitting ability and prediction accuracy. Common methods for calculating correlation include the CORREL function, Pearson analysis, and Spearman analysis. When sample data follow the normal distribution, the correlation coefficient is usually calculated using the CORREL function or Pearson analysis, whereas Spearman analysis is used for non-normally distributed data. This study conducted a normality test on remote sensing reflectance data and obtained significant $p$ values of less than 0.05 for 31 samples, with absolute kurtosis values of less than 10 and absolute skewness values of less than 3. This result indicates that the sample data basically conformed to the normal distribution. On the basis of the three different correlation calculation methods, linear models for Chl-a concentration inversion were established, as shown in Table 6.

**Table 6.** Linear model inversion accuracy based on different correlation analysis methods.

| Analysis Method | Independent Variable | $R^2$ | RMSE | MRE |
|---|---|---|---|---|
| Pearson | B24 | 0.31 | 6.66 | 90.62 |
| Spearman | B27 | 0.35 | 6.43 | 89.70 |

When Pearson correlation analysis was used, the single band with the highest correlation was B27, with a correlation coefficient of 0.58. The linear model fitted on this basis had a coefficient of determination of 0.35. When Spearman correlation analysis was used, the highest correlation coefficient was 0.47 for the single band B24, and the coefficient of determination of the fitted linear model was 0.31. However, the linear inversion model obtained at this point had a much lower inversion accuracy than the BP neural network model. Therefore, when the correct correlation analysis method or neural network model is used instead of the linear models, the accuracy of the Chl-a concentration inversion can be improved to a certain extent.

*4.2. BP Neural Network Model*

This study demonstrated that a direct relationship existed between the fitting ability and prediction accuracy of the BP neural network model by determining the optimal number of hidden layer neurons. When the BP neural network model with only one hidden layer node was selected, the coefficient of determination of the model was only 0.69, and the average relative error between the predicted and measured values of the Chl-a concentration was as high as 35.82%. Therefore, when the number of hidden layer neurons was 12, the fitting ability and prediction accuracy of the model were the best.

This study also showed that, compared with References [1,7,10], the BP neural network model constructed in this work improved the inversion accuracy, especially because the average concentration of Chl-a in the western waters of Hong Kong was much lower than the water quality concentrations in References [1,7,10]. The root mean square error of the results obtained was 9.69%, which is lower than the 10.69%, 16.5%, and 45% in the references. This result was obtained because the correct band ratio was selected for the input layer, and the optimal number of hidden layer neurons was chosen in this study. The band ratios for each of the five terms as input factors are B8/B13, B7/B13, B8/B15, B7/B15, and B8/B16.

This study showed that in the same research area and based on the same satellite remote sensing data, the constructed BP neural network model has the ability to perform joint modelling across seasons. The seasonal distribution characteristics of the predicted values of the BP neural network model established in this study are consistent with the distribution characteristics in other studies on the area [8,9]. On the one hand, the same hyperspectral remote sensing data provide the possibility of cross-seasonal modelling. On the other hand, the variation characteristics of seasonal water quality concentration data may become increasingly obvious, so constructing the relationship between water quality concentration and remote sensing reflectance is easy.

**5. Conclusions**

In this study, a BP neural network method was used to predict the concentration of Chl-a at 42 monitoring points in July and November 2020. The evaluation criteria $R^2$, ARV, and MSE showed that the model had the best fitting effect when the number of hidden layer neurons was 12. The predicted Chl-a concentrations were calculated for each validation sample and compared with the actual values. The results showed that the RMSE and MRE between the predicted and actual values of Chl-a concentrations were 2.12 µg/L and 9.66%, respectively. The model was able to accurately reflect the changes in Chl-a concentrations in different seasons in the study area and their spatial distribution. On the basis of the predicted values, Chl-a concentration maps were drawn for different seasons to analyse the spatial distribution of Chl-a in the study area. The results demonstrated that the BP

neural network model could effectively predict the concentration of Chl-a in the western waters of Hong Kong and conformed to the temporal and spatial distribution patterns of Chl-a in this area. Therefore, this method has potential applications in ocean water quality monitoring and the spatiotemporal analysis of water quality concentrations in the future.

**Author Contributions:** Conceptualization, W.-D.Z. and Y.-X.K.; methodology, Z.-G.L. and Y.-X.K.; validation, W.-D.Z. and N.-Y.H.; formal analysis, W.-D.Z. and Z.-G.L.; resources, Z.-G.Q.; data curation, Z.-G.Q.; writing—original draft preparation, Y.-X.K.; writing—review and editing, W.-D.Z.; visualization, Y.-X.K.; supervision, Z.-G.L.; project administration, N.-Y.H.; funding acquisition, W.-D.Z. and N.-Y.H. All authors have read and agreed to the published version of the manuscript.

**Funding:** This work was supported by the National Key R&D Program of China (2016YFC1400904) and the scientific innovation program project by the Shanghai Committee of Science and Technology (Grant No. 20dz1206501).

**Institutional Review Board Statement:** Not applicable.

**Informed Consent Statement:** Informed consent was obtained from all subjects involved in the study.

**Data Availability Statement:** Data are available from the corresponding author upon request and subject to the Human Subjects protocol restrictions.

**Acknowledgments:** Thank you to the measured concentration data of chlorophyll-a provided by the Environmental Protection Department of the Hong Kong Special Administrative Region; thank you to The Zhuhai No. 1 Hyplateral Satellite Image provided by Zhuhai Orbita Aerospace Science and Technology Limited Company.

**Conflicts of Interest:** The authors declare no conflict of interest.

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
