# Peer review of "Prediction and Analysis of Chlorophyll-a Concentration in the Western Waters of Hong Kong Based on BP Neural Network"

_sustainability, doi:10.3390/su151310441_

Round 1

Author Response

Dear reviewers, experts:

Greetings! First, thank you for your patience and careful guidance. Thank you for your review and insightful summary of the content of this article and for your approval of the content and methodology.

Finally, thank you again and the reviewers for your valuable comments! And at the same time, I hope that if you find any deficiencies again during the review process, please inform us in time, and I will make further amendments to the suggestions.

Sincerely,

Kong.yuxiang

Reviewer 2 Report

In my opinion, the article is at an early stage and does not yet meet the minimum requirements for its publication.

Each of the sections requires significant changes and corrections.

1. Abstract. I recommend that authors redo the abstract after making changes to the body of the article. An abstract is a should give a pertinent overview of the work and should be an objective representation of the article. I recommend briefly describe the background, the main methods, summarize the article’s main results, indicate the main conclusions.

2. The introduction section should be strengthened by clearly identifying the research hypothesis, research questions, objectives, motivations, and novelty of the research. It will help the reader to understand the context and purpose of the study better. Also, the section lacks justification for using NN to solve this type of problem.

Methods section.

3. The authors poorly described the process of creating, preparing and using input data. The process of creating a dataset (input data, output data), cleaning, data preparation is not described.

4. I recommend that the authors show an example of the input and output data in the form of a table.

5. Section 2.4. is called "Spearman Correlation Analysis", the authors describe Pearson correlation.

6. Line 223 "... B1–B5 from top to bottom" in fig. 6. input factors are signed differently.

7. Line 242. ".. single-hidden-layer three-layer BP neural network ...". Please explain.

8. Line 245. I suggest the authors use the classic "neurons" instead of nodes.

9. In my opinion, section 2.6 and the following are already results.

10. Fig. 8. not clear why B1 and B2 are on top? And B2 is directly related to the output variable. In addition, the authors have already shown the Neural Network Architecture in Figure 6.

11. The results are poorly described. The authors simply put the reader in front of the fact, without explaining their actions and calculations step by step. The remark concerns the table. 4, fig. 10 and 11.

12. Line 319. The indicators have not been described before.

13. Lines 390-395. It is necessary to conduct a numerical comparison of the obtained results of this study and previous works.

Author Response

Dear reviewers, experts:

Greetings! First, thank you for your patience and careful guidance. In response to the review comments of the reviewer, I combined the original article to answer the questions and form the text. In order to facilitate the experts to re-examine, I will carry out the various problems pointed out by the experts. Provided a one-to-one answer and marked the key content in red.

Finally, thank you again and the reviewers for your valuable comments! And at the same time, I hope that if you find any deficiencies again during the review process, please inform us in time, and I will make further amendments to the suggestions.

Sincerely,

Kong.yuxiang

Question and Answer

  1. I would suggest that the author redo the abstract after making changes to the body of the article. The abstract is supposed to give a pertinent overview of the work and should be an objective presentation of the article. I would suggest a brief description of the background, the main methods, a summary of the main results of the article and an indication of the main conclusions.

Answer:

Thank you very much for your suggestion. Specific changes have been made to the abstract section (lines 16 - 39) in response to your comments, including an introduction to the background of the remote sensing-based water quality study, the basic methodology and process of the article study, the main results and conclusions.

  1. The introductory section should be strengthened by clearly stating the research hypothesis, the research question, the purpose of the study, the motivation and the novelty of the study. This will help the reader to better understand the context and purpose of the study. Also, the section lacks justification for the use of NN to address this type of question.

Answer:

Thank you for your comments and changes have been made to the introductory section to incorporate your comments. The purpose and motivation of this study is to achieve inter-seasonal high-precision inversion of chlorophyll a concentrations in the study area using hyperspectral image data for the complex water quality environment of offshore waters; the specific research questions and their processes include water quality data and remote sensing image data acquisition, remote sensing image pre-processing, correlation calculation, BP neural network model building and accuracy checking, and spatial and temporal distribution analysis of chlorophyll a concentrations in the study area; The rationale for using the BP neural network model for chlorophyll a concentration inversion is that the chlorophyll a concentration varies greatly across seasons, the non-linear characteristics between data are enhanced, and the BP neural network model can better adapt to the complex non-linear characteristics between data. (lines 82 - 97)

  1. The authors describe the process of creating, preparing and using the input data poorly. The process of creating the data set (input data, output data), cleaning, and data preparation is not described.

Answer:

The input data of this paper is the band reflectance ratio of Zhuhai-1 hyperspectral remote sensing data, which is modified with your comments as follows: the process of data creation and preparation is image application download and image pre-processing, the correlation between chlorophyll a concentration and reflectance is calculated after extracting each band reflectance, and the five sets of reflectance data with correlation coefficients higher than 0.7 are extracted as model input data; the data are used The five reflectance data were used as input factors to build the BP neural network model.

  1. I suggest that the authors show examples of input and output data in the form of a table.

Answer:

Thank you for your suggestion. A table of the input data has been added (table4); the output data is shown in table 5.

  1. Section 2.4 is called "Spearman Correlation Analysis" and the authors describe Pearson's correlation.

Answer:

Thank you very much for pointing out the error, and the appropriate correction has been made (line 206)

  1. line 223 in Figure 6 "..... .B1-B5 from top to bottom", the sign of the input factor is different.

Answer:

Thank you for your comments. The input factors to the model are ratios between wave reflectances, so the five input factors can be understood as purely numerical variables and do not carry units. The five input factors are B8/B13, B7/B13, B8/B15, B7/B15 and B8/B16 and their specific values are shown in Table 4.

  1. Line 242." ...... Single-layer hidden three-layer BP neural network. ......" . Please explain.

Answer:

A single-layer hidden three-layer BP neural network is an artificial neural network with a three-layer structure, including an input layer, a hidden layer and an output layer. The input layer is the starting point of the network and is used to receive the input data, with each input node representing a feature of the input data. The hidden layer is located between the input and output layers, and its role is to process and transform the input data, extract the key features from the data and pass these features to the output layer. Each node in the hidden layer is connected to all the nodes in the input layer and is weighted to adjust the degree of influence of the input data. The output layer is the end point of the network, which performs the final calculations and predictions based on the features passed through the hidden layer, and outputs the results of the network.

  1. line 245. I would suggest that the authors use the classical "neurons" rather than nodes.

Answer:

Thank you very much for your suggestion, and the corresponding changes have been made (line 259), and the other nine identical errors in the text have been corrected.

  1. It seems to me that section 2.6 and the following are already results.

Answer:

Thank you very much for your comments and sections 2.6 and 2.7 have been incorporated into the section in Chapter 3 (Results and Analysis).

  1. figure 8. it is not clear why B1 and B2 are on top? and B2 is directly related to the output variable. Also, the authors have shown the neural network architecture in Figure 6.

Answer:

Thank you for your comments. B1 and B2 in Figure 8 are not input variables, but bias nodes. The bias nodes are designed to describe features that are not present in the training data and are biased differently for each node in the next layer with different weights; the bias nodes themselves do not carry fixed weight parameters. The core idea of the BP neural network in this paper is that the input of a layer is multiplied by the corresponding weights plus a bias, and the output after substitution into the activation function is again the input of the next layer.

The neural network diagram in Figure 6 is based on an illustration of the principle of a BP neural network and thus shows its structure. Figure 8 shows the basic operational structure of model 12 after determining the number of neurons in the hidden layer. Thank you again for your comments.

  1. The results are poorly described. The authors simply put the reader in front of the facts without explaining their actions and calculations step by step. The comment relates to Table. 4, Figures 10 and 11.

Answer:

Thank you for your suggestion. Changes have been made to address the results section of the article, which describes the source of the results and the process of obtaining them.

  1. line 319. These indicators have not been described previously.

Answer:

Thank you for your comment. The three accuracy evaluation metrics here are relative error (RE), root mean square error (RMSE), and mean relative error (MRE). The formulae for their calculation are given in the text (line 341)

13. lines 390-395. It is necessary to make a numerical comparison between the results obtained in this study and previous work.

Answer:

Thank you for your comments. A comparison of the precision of the results with those of the references has been added to the text paragraph and the root mean square error of the results obtained is 9.69%, which is lower than the 10.69%, 16.5% and 45% in Ref. (line 413)

Reviewer 3 Report

General: 

An interesting and novel approach to a difficult problem. Paper is well-written and the work is well done. 

The difficulty for retrieving chlorophyll-a in coastal waters is that many of the standard algorithms fail due to optical materials in the water other than chlorophyll that change the spectral shape of the reflectance signal. Authors might consider discussing this and then comparing their results to one of these standard chlorophyll algorithms. This is only a suggestion. 

Also, it is well -known that chlorophyll will impact the blue to green bands and have little impact on the remaining bands. Thus, looking at correlations with the higher bands in Table 2 is not likely to be productive. 

Most investigators in this field would want to know exactly what bands, or combination of bands—and exactly how they were combined (ratios? Log-transform? What?) – were used as input for the successful models. 

Specific 

Line 41 – Zooplankton do not contain chlorophyll-a pigment

Figure 11- It is often customary to log-transform the chl-a values and display using a color table, this permits more details to be apparent in the visualization

What is the exact input for model 12? Is it the raw bands, the ratio of bands, or some combination thereof? Other investigators should be able to reproduce your results. More specificity is required here. 

Entries in Table 4 appear to be missing. 

Line 394 – “the correct band combination” – please specify what that band combination is. . . 

If authors have developed an algorithm that may be used in coastal waters -- then they need to publish exactly what it is - including supplemental materials -- so the rest of the scientific community can use it and compare it to what they are doing. 

If authors refuse to publish the details of their algorithm, then we cannot reproduce their results. 

Author Response

Dear reviewers, experts:

Greetings! First, thank you for your patience and careful guidance. In response to the review comments of the reviewer, I combined the original article to answer the questions and form the text. In order to facilitate the experts to re-examine, I will carry out the various problems pointed out by the experts. Provided a one-to-one answer and marked the key content in red.

Finally, thank you again and the reviewers for your valuable comments! And at the same time, I hope that if you find any deficiencies again during the review process, please inform us in time, and I will make further amendments to the suggestions.

Sincerely,

Kong.yuxiang

Question and Answer

  1. Most researchers in this field want to know exactly what bands, or combinations of bands, and exactly how they are combined (ratios? logarithmic conversion?) . Why calculate the correlation between some high bands and chlorophyll a concentration.

Answer:

Thank you very much for your correction. The selection of sensitive bands in this paper relies on the correlation coefficient between band reflectance and chlorophyll a concentration, and the calculations used in this paper include band ratios and logarithmic transformations. (Lines 220-225) The input factors with the highest correlation coefficients were all selected as the ratio of band reflectance, and the five input factors were B8/B13, B7/B13, B8/B15, B7/B15, and B8/B16. (Figure 6)

For the higher bands the use was made in consideration of the possible stochastic nature of the combined results between reflectance values, i.e. it is possible that the ratio of a particular higher band to the blue-green band has a higher correlation with chlorophyll a concentration.

  1. line 41 - zooplankton do not contain chlorophyll a pigment

Answer:

Thank you very much for your correction, changes have been made to address the content here. (line 48)

3, Figure 11 - The usual convention is to log-transform chl-a values and display them using a coloured table, which allows more detail to be shown in the visualisation.

Answer:

Thank you very much for your correction. Considering that the direct use of chlorophyll a concentration values to indicate the concentration distribution facilitates a more intuitive understanding of the chlorophyll a concentration values and their water quality class in a particular water area, and considering the large differences in chlorophyll a concentration values in space and time, the spatial and temporal distribution characteristics of the chlorophyll a concentration values can be intuitively understood in the concentration distribution map here, therefore no logarithmic calculation of the concentration values was performed and The values are therefore not logarithmically calculated and displayed. Thank you again for your suggestion.

  1. What is the exact input to model 12? Is it the raw bands, the ratio of the bands, or some combination of them? Other researchers should be able to reproduce your results. More specific instructions are needed here.

Answer:

The five input factors to model 12 are the band reflectance ratios (B8/B13, B7/B13, B8/B15, B7/B15, B8/B16), the number of neurons in the implicit layer is 12, and the output layer is the chlorophyll a concentration value. Corresponding changes have been made in the article (lines 303 - 305).

  1. The entry in Table 4 seems to be missing.

Answer:

The last two metrics in Table 4 are root mean square error (RMSE) and mean relative error (MRE), which are used to represent the overall error between the predicted and measured chlorophyll a concentration values, so there is only a single root mean square error and mean relative error for the concentration values at the eight sites, and no data are missing.

  1. line 394 - 'correct band combination' - please state what this band combination is...

Answer:

The band combination is the five band reflectance ratios used in this paper - B8/B13, B7/B13, B8/B15, B7/B15, B8/B16. (Fig. 6) A change has been made here to change the band combination to a band ratio, adding specific input factors (lines 416)

  1. If the authors have developed an algorithm that can be used in coastal waters - then they need to publish exactly what it is - including supplementary material - so that other members of the scientific community can use it and compare it with what they are doing.

Answer:

The modelling method used in this paper is a BP neural network model, implemented in RStudio. The specific code is shown below and the parameters in the code may vary due to changes in the data. However, as the study area is coastal waters, the spatial and temporal variation rate of each water quality parameter is relatively fast, so it may lead to relatively poor generalisation of the model.( This is the basic code, which needs to be adjusted according to the experimental parameters and the actual situation)

library(nnet)

library(NeuralNetTools)

mydata=read.csv("jianmo.csv", sep=",",header = T,stringsAsFactors = FALSE)

names(mydata)

MSE=matrix(data= NA,ncol= 20,nrow= 100)

ARV=matrix(data= NA,ncol= 20,nrow= 100)

r_square=matrix(data = NA,ncol = 20,nrow = 100)

for(i in 1:100)

{

  index = sample(1:nrow(mydata),0.7*nrow(mydata),replace = F)

  train = mydata[index,]

  test = mydata[-index,]

  model1  <-  nnet(NChl_a~B1+B2+B3+B4+B5, train, size = 1, decay = 0.01, maxit = 1000, linout = T, trace = F)

  model2  <-  nnet(NChl_a~B1+B2+B3+B4+B5, train, size = 2, decay = 0.01, maxit = 1000, linout = T, trace = F)

  model3  <-  nnet(NChl_a~B1+B2+B3+B4+B5, train, size = 3, decay = 0.01, maxit = 1000, linout = T, trace = F)

  model4  <-  nnet(NChl_a~B1+B2+B3+B4+B5, train, size = 4, decay = 0.01, maxit = 1000, linout = T, trace = F)

  model5  <-  nnet(NChl_a~B1+B2+B3+B4+B5, train, size = 5, decay = 0.01, maxit = 1000, linout = T, trace = F)

  model6  <-  nnet(NChl_a~B1+B2+B3+B4+B5, train, size = 6, decay = 0.01, maxit = 1000, linout = T, trace = F)

  model7  <-  nnet(NChl_a~B1+B2+B3+B4+B5, train, size = 7, decay = 0.01, maxit = 1000, linout = T, trace = F)

  model8  <-  nnet(NChl_a~B1+B2+B3+B4+B5, train, size = 8, decay = 0.01, maxit = 1000, linout = T, trace = F)

  model9  <-  nnet(NChl_a~B1+B2+B3+B4+B5, train, size = 9, decay = 0.01, maxit = 1000, linout = T, trace = F)

  model10  <-  nnet(NChl_a~B1+B2+B3+B4+B5, train, size = 10, decay = 0.01, maxit = 1000, linout = T, trace = F)

  model11  <-  nnet(NChl_a~B1+B2+B3+B4+B5, train, size = 11, decay = 0.01, maxit = 1000, linout = T, trace = F)

  model12  <-  nnet(NChl_a~B1+B2+B3+B4+B5, train, size = 12, decay = 0.01, maxit = 1000, linout = T, trace = F)

  model13  <-  nnet(NChl_a~B1+B2+B3+B4+B5, train, size = 13, decay = 0.01, maxit = 1000, linout = T, trace = F)

  model14  <-  nnet(NChl_a~B1+B2+B3+B4+B5, train, size = 14, decay = 0.01, maxit = 1000, linout = T, trace = F)

  model15  <-  nnet(NChl_a~B1+B2+B3+B4+B5, train, size = 15, decay = 0.01, maxit = 1000, linout = T, trace = F)

  model16  <-  nnet(NChl_a~B1+B2+B3+B4+B5, train, size = 16, decay = 0.01, maxit = 1000, linout = T, trace = F)

  model17  <-  nnet(NChl_a~B1+B2+B3+B4+B5, train, size = 17, decay = 0.01, maxit = 1000, linout = T, trace = F)

  model18  <-  nnet(NChl_a~B1+B2+B3+B4+B5, train, size = 18, decay = 0.01, maxit = 1000, linout = T, trace = F)

  model19  <-  nnet(NChl_a~B1+B2+B3+B4+B5, train, size = 19, decay = 0.01, maxit = 1000, linout = T, trace = F)

  model20  <-  nnet(NChl_a~B1+B2+B3+B4+B5, train, size = 20, decay = 0.01, maxit = 1000, linout = T, trace = F)

  predict1=predict(model1,test)

  predict2=predict(model2,test)

  predict3=predict(model3,test)

  predict4=predict(model4,test)

  predict5=predict(model5,test)

  predict6=predict(model6,test)

  predict7=predict(model7,test)

  predict8=predict(model8,test)

  predict9=predict(model9,test)

  predict10=predict(model10,test)

  predict11=predict(model11,test)

  predict12=predict(model12,test)

  predict13=predict(model13,test)

  predict14=predict(model14,test)

  predict15=predict(model15,test)

  predict16=predict(model16,test)

  predict17=predict(model17,test)

  predict18=predict(model18,test)

  predict19=predict(model19,test)

  predict20=predict(model20,test)

  MSE_Model1=sum((predict1-test$NChl_a)^2)/nrow(test)

  MSE_Model2=sum((predict2-test$NChl_a)^2)/nrow(test)

  MSE_Model3=sum((predict3-test$NChl_a)^2)/nrow(test)

  MSE_Model4=sum((predict4-test$NChl_a)^2)/nrow(test)

  MSE_Model5=sum((predict5-test$NChl_a)^2)/nrow(test)

  MSE_Model6=sum((predict6-test$NChl_a)^2)/nrow(test)

  MSE_Model7=sum((predict7-test$NChl_a)^2)/nrow(test)

  MSE_Model8=sum((predict8-test$NChl_a)^2)/nrow(test)

  MSE_Model9=sum((predict9-test$NChl_a)^2)/nrow(test)

  MSE_Model10=sum((predict10-test$NChl_a)^2)/nrow(test)

  MSE_Model11=sum((predict11-test$NChl_a)^2)/nrow(test)

  MSE_Model12=sum((predict12-test$NChl_a)^2)/nrow(test)

  MSE_Model13=sum((predict13-test$NChl_a)^2)/nrow(test)

  MSE_Model14=sum((predict14-test$NChl_a)^2)/nrow(test)

  MSE_Model15=sum((predict15-test$NChl_a)^2)/nrow(test)

  MSE_Model16=sum((predict16-test$NChl_a)^2)/nrow(test)

  MSE_Model17=sum((predict17-test$NChl_a)^2)/nrow(test)

  MSE_Model18=sum((predict18-test$NChl_a)^2)/nrow(test)

  MSE_Model19=sum((predict19-test$NChl_a)^2)/nrow(test)

  MSE_Model20=sum((predict20-test$NChl_a)^2)/nrow(test)

  MSE[i,1]=MSE_Model1

  MSE[i,2]=MSE_Model2

  MSE[i,3]=MSE_Model3

  MSE[i,4]=MSE_Model4

  MSE[i,5]=MSE_Model5

  MSE[i,6]=MSE_Model6 

  MSE[i,7]=MSE_Model7 

  MSE[i,8]=MSE_Model8

  MSE[i,9]=MSE_Model9

  MSE[i,10]=MSE_Model10

  MSE[i,11]=MSE_Model11

  MSE[i,12]=MSE_Model12

  MSE[i,13]=MSE_Model13

  MSE[i,14]=MSE_Model14 

  MSE[i,15]=MSE_Model15 

  MSE[i,16]=MSE_Model16

  MSE[i,17]=MSE_Model17

  MSE[i,18]=MSE_Model18

  MSE[i,19]=MSE_Model19

  MSE[i,20]=MSE_Model20

  ARV_Model1 =sum((predict1-test$NChl_a)^2)/sum((mean(test$NChl_a)-test$NChl_a)^2)

  ARV_Model2 =sum((predict2-test$NChl_a)^2)/sum((mean(test$NChl_a)-test$NChl_a)^2)

  ARV_Model3 =sum((predict3-test$NChl_a)^2)/sum((mean(test$NChl_a)-test$NChl_a)^2)

  ARV_Model4 =sum((predict4-test$NChl_a)^2)/sum((mean(test$NChl_a)-test$NChl_a)^2)

  ARV_Model5 =sum((predict5-test$NChl_a)^2)/sum((mean(test$NChl_a)-test$NChl_a)^2)

  ARV_Model6 =sum((predict6-test$NChl_a)^2)/sum((mean(test$NChl_a)-test$NChl_a)^2)

  ARV_Model7 =sum((predict7-test$NChl_a)^2)/sum((mean(test$NChl_a)-test$NChl_a)^2)

  ARV_Model8 =sum((predict8-test$NChl_a)^2)/sum((mean(test$NChl_a)-test$NChl_a)^2)

  ARV_Model9 =sum((predict9-test$NChl_a)^2)/sum((mean(test$NChl_a)-test$NChl_a)^2)

  ARV_Model10 =sum((predict10-test$NChl_a)^2)/sum((mean(test$NChl_a)-test$NChl_a)^2)

  ARV_Model11 =sum((predict11-test$NChl_a)^2)/sum((mean(test$NChl_a)-test$NChl_a)^2)

  ARV_Model12 =sum((predict12-test$NChl_a)^2)/sum((mean(test$NChl_a)-test$NChl_a)^2)

  ARV_Model13 =sum((predict13-test$NChl_a)^2)/sum((mean(test$NChl_a)-test$NChl_a)^2)

  ARV_Model14 =sum((predict14-test$NChl_a)^2)/sum((mean(test$NChl_a)-test$NChl_a)^2)

  ARV_Model15 =sum((predict15-test$NChl_a)^2)/sum((mean(test$NChl_a)-test$NChl_a)^2)

  ARV_Model16 =sum((predict16-test$NChl_a)^2)/sum((mean(test$NChl_a)-test$NChl_a)^2)

  ARV_Model17 =sum((predict17-test$NChl_a)^2)/sum((mean(test$NChl_a)-test$NChl_a)^2)

  ARV_Model18 =sum((predict18-test$NChl_a)^2)/sum((mean(test$NChl_a)-test$NChl_a)^2)

  ARV_Model19 =sum((predict19-test$NChl_a)^2)/sum((mean(test$NChl_a)-test$NChl_a)^2)

  ARV_Model20 =sum((predict20-test$NChl_a)^2)/sum((mean(test$NChl_a)-test$NChl_a)^2)

  ARV[i,1] =ARV_Model1

  ARV[i,2] =ARV_Model2

  ARV[i,3] =ARV_Model3

  ARV[i,4] =ARV_Model4

  ARV[i,5] =ARV_Model5

  ARV[i,6] =ARV_Model6 

  ARV[i,7] =ARV_Model7 

  ARV[i,8] =ARV_Model8

  ARV[i,9] =ARV_Model9

  ARV[i,10] =ARV_Model10

  ARV[i,11] =ARV_Model11

  ARV[i,12] =ARV_Model12

  ARV[i,13] =ARV_Model13

  ARV[i,14] =ARV_Model14 

  ARV[i,15] =ARV_Model15 

  ARV[i,16] =ARV_Model16

  ARV[i,17] =ARV_Model17

  ARV[i,18] =ARV_Model18

  ARV[i,19] =ARV_Model19

  ARV[i,20] =ARV_Model20

  lm1 = lm(predict1~test$NChl_a)

  lm2 = lm(predict2~test$NChl_a)

  lm3 = lm(predict3~test$NChl_a)

  lm4 = lm(predict4~test$NChl_a)

  lm5 = lm(predict5~test$NChl_a)

  lm6 = lm(predict6~test$NChl_a)

  lm7 = lm(predict7~test$NChl_a)

  lm8 = lm(predict8~test$NChl_a)

  lm9 = lm(predict9~test$NChl_a)

  lm10 = lm(predict10~test$NChl_a)

  lm11 = lm(predict11~test$NChl_a)

  lm12 = lm(predict12~test$NChl_a)

  lm13 = lm(predict13~test$NChl_a)

  lm14 = lm(predict14~test$NChl_a)

  lm15 = lm(predict15~test$NChl_a)

  lm16 = lm(predict16~test$NChl_a)

  lm17 = lm(predict17~test$NChl_a)

  lm18 = lm(predict18~test$NChl_a)

  lm19 = lm(predict19~test$NChl_a)

  lm20 = lm(predict20~test$NChl_a)

  r_square[i, 1] = summary(lm1)$r.squared

  r_square[i, 2] = summary(lm2)$r.squared

  r_square[i, 3] = summary(lm3)$r.squared

  r_square[i, 4] = summary(lm4)$r.squared

  r_square[i, 5] = summary(lm5)$r.squared

  r_square[i, 6] = summary(lm6)$r.squared

  r_square[i, 7] = summary(lm7)$r.squared

  r_square[i, 8] = summary(lm8)$r.squared

  r_square[i, 9] = summary(lm9)$r.squared

  r_square[i, 10] = summary(lm10)$r.squared

  r_square[i, 11] = summary(lm11)$r.squared

  r_square[i, 12] = summary(lm12)$r.squared

  r_square[i, 13] = summary(lm13)$r.squared

  r_square[i, 14] = summary(lm14)$r.squared

  r_square[i, 15] = summary(lm15)$r.squared

  r_square[i, 16] = summary(lm16)$r.squared

  r_square[i, 17] = summary(lm17)$r.squared

  r_square[i, 18] = summary(lm18)$r.squared

  r_square[i, 19] = summary(lm19)$r.squared

  r_square[i, 20] = summary(lm20)$r.squared

}

boxplot(r_square,

        ylim=c(0,1.0),

        xlab="models",

        ylab="R_square",

        main="",

        col="white")

boxplot(MSE,

        ylim=c(0,0.1),

        xlab="models",

        ylab="MSE",

        main="",

        col="white")

boxplot(ARV,

        ylim=c(0,1.5),

        xlab="models",

        ylab="ARV",

        main="",

        col="white")

jpeg(filename = "ARV600.jpeg",width = 1000,height = 1500, res = 400)

par(mar = c(1, 1, 1, 1), oma = c(2, 2, 1, 1), mai=c(0,0,0,0), tcl = -0.1, mgp = c(2, 0.6, 0), family='serif')

boxplot(ARV,

        ylim=c(0,1.5),

        xlab="models",

        ylab="ARVÖµ",

        col="white")

dev.off()

jpeg(filename = "R^2600.jpeg",width = 1000,height = 1500, res = 400)

par(mar = c(1, 1, 1, 1), oma = c(2, 2, 1, 1), mai=c(0,0,0,0), tcl = -0.1, mgp = c(2, 0.6, 0), family='serif')

boxplot(r_square,

        ylim=c(0,1),

        xlab="models",

        ylab="R²",

        main="",

        col="white")

dev.off()

print  (model12)

jpeg(filename = "Garson.jpeg",width = 2000,height = 1200,res = 500)

garson (model12)

dev.off()

jpeg(filename = "Plotnat.jpeg",width = 5000,height = 4000,res = 600)

plotnet(model12)

dev.off()

predata = read.csv("new.csv",header = T)

names(predata)

pre1=predict(model12,predata)

write.csv(pre1, file='pre.csv', row.names = T)

Round 2

Reviewer 2 Report

I recommend publishing the article in present form.